# Multi-client distributed blind quantum computation with the Qline architecture

Beatrice Polacchi [1], Dominik Leichtle [2], Leonardo Limongi [1], Gonzalo Carvacho [1], Giorgio Milani [1], Nicolò Spagnolo[1], Marc Kaplan [3] ✉, Fabio Sciarrino [1] ✉ & Elham Kashefi [2,4,5] ✉

Universal blind quantum computing allows users with minimal quantum resources to delegate a quantum computation to a remote quantum server, while keeping intrinsically hidden input, algorithm, and outcome. State-of-art experimental demonstrations of such a protocol have only involved one client. However, an increasing number of multi-party algorithms, e.g. federated machine learning, require the collaboration of multiple clients to carry out a given joint computation. In this work, we propose and experimentally demonstrate a lightweight multi-client blind quantum computation protocol based on a recently proposed linear quantum network configuration (Qline). Our protocol originality resides in three main strengths: scalability, since we eliminate the need for each client to have its own trusted source or measurement device, low-loss, by optimizing the orchestration of classical communication between each client and server through fast classical electronic control, and compatibility with distributed architectures while remaining intact even against correlated attacks of server nodes and malicious clients.

Despite the increasing technological progress in the manipulation of many-qubit systems[1–3], providing quantum computing as a service for end-users poses several challenges, including scalability, privacy, and integrity. Indeed, in order to achieve true quantum advantage from emerging devices, they must scale up beyond the current monolithically designed noisy intermediate-scale quantum regime. As of today, the only viable solution being pursued by all qubit platforms is modularity and interconnected architecture, where photonic links are considered the best option. Moreover, it is also clear that quantum machines need to be integrated into cloud services or data centers, allowing multiple clients to connect locally or globally to access these devices. In such a context, the issue of keeping the computation and data protected from malicious parties will be a key challenge for such large-scale adaptation. Notably, photonic links to quantum servers enable the capability of achieving informational security for delegated computing, known as blind quantum computing (BQC), which is not achievable using only classical communication between client and server[4]. Such a protocol builds on the measurement-based model for quantum computation[5,6] that exploits mid-circuit measurements for teleportation-based quantum computing on encrypted quantum states sent to a remote server via a quantum link[7,8]. Over the last decade, many BQC protocols have been proposed[9–25], together with proof-of-concept experimental demonstrations in different settings[26–35]. However, the challenge of multi-client settings has been explored only theoretically due to the high resource requirements of the proposed protocols[36–40].

Yet, a growing number of classical delegated computing tasks require that multiple clients collaborate to carry out a joint function, e.g., federated machine learning tasks[41,42]. Notably, quantum counterparts of such algorithms have been proposed as well[43], including a federated quantum machine learning (QML) protocol based on BQC[44].

[1]Dipartimento di Fisica, Sapienza Università di Roma, P.le Aldo Moro 5, I-00185 Roma, Italy. [2]Laboratoire d'Informatique de Paris 6, CNRS, Sorbonne Université, 75005 Paris, France. [3]VeriQloud, 13 rue Victor Hugo, 92120 Montrouge, France. [4]School of Informatics, University of Edinburgh, 10 Crichton Street, EH8 9AB Edinburgh, UK. [5]National Quantum Computing Centre, Didcot OX11 0QX, UK. ✉e-mail: kaplan@veriqloud.fr; fabio.sciarrino@uniroma1.it; elham.kashefi@lip6.fr

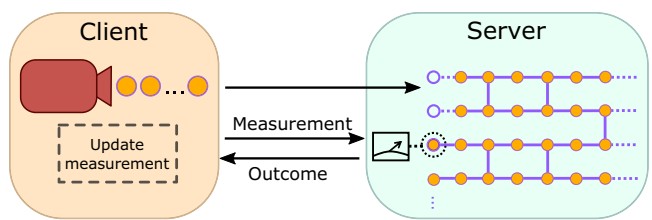

**Fig. 1 | Conceptual scheme of BQC.** In the preparation stage of the BQC protocol, a client randomly prepares $m$ qubits and sends them to a quantum server. The server uses the qubits to form a resource state for the computation. In the measurement stage, $\mathcal{O}(m)$ rounds of classical communication between a client and a server are needed to carry out the computation. At each round, the server measures a qubit in a measurement basis suitably chosen by the client in order to hide the computation details, and it gives back the measurement outcome to the client. Then, the latter decides on the next measurement basis accordingly. Further detail about BQC is given in Supplementary Protocol 1. Figure inspired by ref. 53.

In this work, we propose a modular lightweight distributed architecture for multi-client BQC based on the recently proposed Qline quantum network link configuration[45], that enables scalable client insertion. With such an architecture in mind, we present a tailor-made multi-party BQC protocol such that the clients in the Qline only own trusted single-qubit rotation devices, while the overall protocol still provides privacy for the joint computation performed by several users on the cloud. The model for BQC in which the client only performs single-qubit gates was already proposed in refs. 11,12. However, although our protocol is inspired by the aforementioned works, we emphasize that its security properties cannot be directly derived from those BQC protocols as our multi-client setting is not a mere composition of such previous works. Therefore, in this work, we also provide full security proof, as we do in the section "Security". In the Qline network architecture, the quantum resource is first generated by a potentially untrusted server, then distributed to the clients, such that each client can apply arbitrary single-qubit operations on the incoming qubits and, at the end of the line, measured by a second again potentially untrusted server. An analogous architecture was introduced in ref. 46 for quantum-assisted secret sharing, and later used for various tasks such as quantum key distribution or secure computation[47–49]. The main advantages of such a structure reside in the possibility to integrate it easily into larger-scale networks, its compatibility with key establishment protocols[45], and its low hardware complexity. In order to simplify the resource requirements of the multi-client BQC protocol in refs. 36,40, we show that, within such an architecture, it is enough that each client in the Qline adds a layer of encryption to the flying qubits that will be used as the common key for their later private joint computation on the remote server. Such collaboration may be typical, for instance, of privacy-preserving machine learning algorithms where each client's input data and parameters related to the algorithm should remain private to all parties. To implement it, we employ a fibered photonic platform equipped with genuine measurement adaptivity to enable deterministic computation[5]. Within this setup, we are able to show the blindness and the correctness of the protocol, in both cases where the function to be computed has a classical or a quantum output. Our experimental proof-of-concept demonstrates a two-client scenario that can be easily extended to larger and more complex quantum networks featuring any number of clients at arbitrary distances.

## Results

In this section, we describe in detail the protocol proposed and successfully implemented in this work. It is built on the theoretical premises in refs. 36,40, and tailored to a Qline architecture[45]. Differently from the single client original protocol of ref. 7 depicted in Fig. 1, the results of refs. 36,40 enable multi-client BQC by exploiting secure

multi-party computation (SMPC), whose aim is to allow several users to collaboratively compute a joint function on their private data. The classical SMPC functionality enables coordination of the parties in a delegated quantum computing task, such that the full computation details are blind not only to the server but also to potentially dishonest clients that collude with it. However, implementing such functionality would need additional rounds of classical communication among the clients and server, during which it would be unfeasible to coherently store the quantum state. Therefore, to optimize the storage time, in our implementation, we substitute classical SMPC with a trusted third party (TTP) acting as an orchestrator and securing the communication between the clients and the server reducing the number of rounds, while the blindness of the protocol is still proven against any strict subset of colluding malicious adversaries. In this way, the quantum state needs to be stored for significantly shorter times than if using full classical SMPC, thus enabling our proof-of-concept experimental demonstration of a two-client BQC. Our solution represents a trade-off between hardware trust assumptions and time latency between rounds. However, the trust assumption on the TTP can be dropped without modifying our scheme, by simply replacing it with classical SMPC, if we run the computation on a quantum server equipped with quantum memory. To motivate our experimental design, this section is divided into two parts: the first one is devoted to the description of a two-client example, which we implemented experimentally demonstrating the key building blocks that are required for a fully scalable solution. In the second part, we present the extension to the $n$-client case and universal quantum computing resources.

### The two-client protocol

Consider Alice and Bob wish to run a joint computation on a remote server. Alice has two private classical bits of information, $x_1$ and $x_2$, while Bob has private gate parameters $\phi_1$ and $\phi_2$, chosen from the set $\mathcal{A} = \{0, \pi/4, 2\pi/4, \dots 7\pi/4\}$, and the target joint circuit is

$$\left(M^X \otimes M^X\right)\left(R_z(\phi_1) \otimes R_z(\phi_2)\right) CZ_{12}\left(Z^{x_1} \otimes Z^{x_2}\right)(|+\rangle \otimes |+\rangle)$$

as shown in Fig. 2a. This is a typical building block of any large-scale privacy-preserving QML, such as the one proposed in ref. 43. Indeed, the distribution and the size of the input data are flexible, making this protocol suitable even for federated machine-learning tasks. For example, each client $C_j$ could provide one measurement angle $\phi_i^{C_j}$ for each qubit $i$, and one classical bit $x_i^{C_j}$. In this case, the cumulative private measurement angle applied to the $i$-th qubit will be $\phi_i = \sum_j \phi_i^{C_j}$, still in the set $\mathcal{A}$, while the initial encoding of classical data will be described by the operator $Z^{\bigoplus_j x_i^{C_j}}$. Also, not all clients are required to provide input data for each qubit. In what follows, we demonstrate the steps to make the above joint computation both distributed and secure as shown in Fig. 2b.

An untrusted source of maximally entangled bipartite states, $S_1$, distributes two-qubit states along two quantum channels, of the form:

$$|\psi\rangle = \frac{1}{2}(|00\rangle + |01\rangle + |10\rangle - |11\rangle) \tag{1}$$

Alice receives the two qubits, and applies single-qubit $z$-rotations of angles $\theta_i^A$ to them, randomly chosen from the set $\mathcal{A}$. This will hide (via quantum one-time padding) her classical input data which would be encoded on these qubits via $Z^{x_i}$ operations. Moreover, she chooses two random bits $r_1^A, r_2^A$ that will later hide the outcome of the computation. She communicates her secret parameters to the TTP. She then sends her two encrypted qubits to the second client, Bob, who applies further random $\theta_1^B, \theta_2^B$ $z$-rotations, again chosen from the set $\mathcal{A}$ to one-time pad his private algorithm parameter $\phi_1, \phi_2$. Bob also

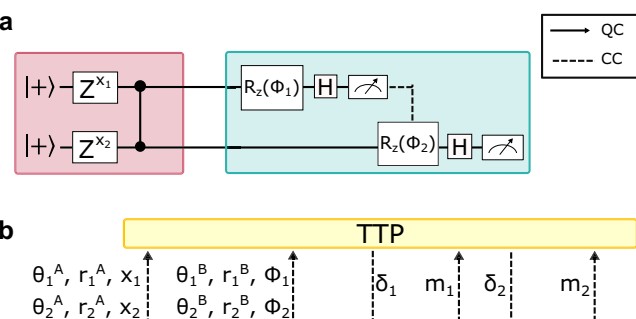

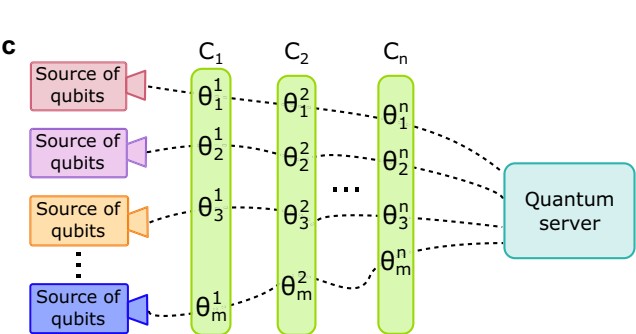

**Fig. 2 | Conceptual scheme of the two-client BQC and distributed quantum computing over a Qline architecture. a** The desired joint quantum circuit computation where $x_1$ and $x_2$ are Alice's private input data and $\phi_1$ and $\phi_2$ the private angles of Bob's algorithm. **b** The same computation of the circuit presented in **a** is encrypted to preserve the privacy of each party's information. In our two-client BQC protocol, a quantum channel connects a source of bipartite quantum states to two clients disposed along the Qline. Each client chooses their secret parameters $\theta_i^j, r_i^j, x_i, \phi_i$, for $i = 1, 2$ and $j = A, B$, and applies $z$-rotations to both qubits. All secret parameters and measurement outcomes pass through a TTP to compute the transformed measurement bases $\delta_i$ and the corrected outcomes. At the end of the line, the quantum state is sent to server $S_2$ to carry out the desired computation, which is carried out through two rounds of classical communication between clients, the TTP, and server $S_2$. **c** Generalization of our architecture to $m$ Qlines with $n$ clients distributed along them. More in general, in a Qline architecture, $m$ independent quantum state sources distribute single qubits to $n$ clients, and each client applies random rotations to all qubits. At the end of the line, a powerful quantum server employs the received qubits to generate the resource state for the computation and calculate a joint function.

chooses two random bits $r_1^B, r_2^B$ for the encryption of the output as well. He then communicates his secret parameters to the TTP. From now on, we will use the following definitions: $\theta_i = \theta_i^A + \theta_i^B$ and $r_i = r_i^A \oplus r_i^B$. The resulting quantum state at this stage is the following:

$$|\psi\rangle = \frac{1}{2}\left(|00\rangle + e^{i\theta_2}|01\rangle + e^{i\theta_1}|10\rangle - e^{i(\theta_1+\theta_2)}|11\rangle\right) \qquad (2)$$

This state is then sent to server $S_2$. From now on, the clients and $S_2$ only communicate classically, through the TTP. The protocol requires two rounds, one for each qubit to be measured. The blind measurement angle $\delta_i$ at the $i$-th round, for $i = 1, 2$, is computed by the TTP according to the formula:

$$\delta_i = \theta_i + x_i\pi + (-1)^{m_{(i-1)}^{true}}\phi_i + r_i\pi \qquad (3)$$

where $m_0^{true} = 0$ and $m_1^{true} = m_1 \oplus r_1$. Analogously to the first measurement outcome, the outcome of the second measurement is decrypted according to the formula: $m_2^{true} = m_2 \oplus r_2$ before giving it

back to the clients. A slight change in the protocol is needed in the case where a quantum function is computed, i.e., the output qubit must be prepared by the clients in the state $|+\rangle$[7]. The correctness of the protocol is straightforward and can be directly obtained from refs. 7,10,40. However, the security proof is more subtle compared to previous works that were based on trusted state preparation for each client. We first present the generalization of our protocol and then provide the full proof of security that is applicable to this special case as well.

## Generalization to the multi-client scenario

In this section, we generalize our protocol to a scenario where $n$ clients want to perform a joint computation on a possibly larger resource state. Note that, here, we treat a more general case with respect to the previous one, hence the security proof we provide, which assumes a fully quantum server and independent and arbitrarily distant qubit sources, also holds for the implemented protocol. Blindness should be guaranteed for any single honest client. We consider the target computation to be defined as a measurement pattern[5] by the measurement angles $(\phi_v)_{v \in V}$, where $v \in V$ ranges over all qubits in the resource graph state. These angles can be fixed and publicly known, or jointly input by any subset of clients. In the latter case, blindness holds for the measurement angles as well. The input qubits $I \subset V$ are partitioned into sets $(I_j)_{j \in \{1,\dots,n\}}$, where $I_j$ belongs to the $j$-th client who has the bit string $x_j \in \{0,1\}^{|I_j|}$ as input. To keep each client's input blind, it is required that each qubit in the resource state travels once along the Qline and accumulates random rotations by all clients, as depicted in Fig. 2c. In this way, a resource state on $m = |V|$ qubits would require $m$ Qlines. However, this process may be (partially) parallelized, in the sense that multiple qubits can be sent along the Qline at once—as long as the clients can perform the necessary rotations in parallel as well. After the server receives the qubits that have passed through the Qline, it follows a standard execution of the BQC protocol (Supplementary Protocol 1). As all communication from this point forward is entirely classical, the TTP orchestrates the remainder of the protocol by governing the instructions to the server. A detailed description of the full multi-client protocol on scalable resource states is given in the protocol in Box 1.

We also point out that the orchestration of the protocol in Box 1 must be performed by an entirely classical trusted third party that secures the secret parameters input by the clients as well as the measurement outcomes, which should not be leaked. To remove all trust assumptions on this party, it can be replaced with any composably secure classical SMPC protocol that is performed by the clients and the server. Moreover, much of the calculations that the TTP needs to perform, including the sampling of random coins and the evaluation of the formulae for the corrected measurement angles, can be done in a classical pre-processing step. The computation during the quantum phase then boils down to the choice of one of two possible measurement angles based on the previously reported measurement outcomes. Finally, it is worth mentioning that the requirement that every client has access to each Qline can be weakened when accepting stronger trust assumptions. Blindness holds as long as there exists at least one honest client along each Qline. Therefore, even if not every client participates at each Qline, blindness is still guaranteed if we restrict the adversarial patterns to not corrupt all clients along any Qline at once. In the concrete case of our experiment, the graph $G$ is a two-qubit cluster state, that is, $V = \{1, 2\}$. Both qubits are Alice's input qubits, so $I_1 = I = V$, while Bob has no input qubits ($I_2 = \emptyset$) but chooses the measurement angles $(\phi_1, \phi_2)$. We consider two different computations: in the first case, we interpret both measurement outcomes as the classical output of the computation, while in the second case, we consider the second qubit to be the quantum output of the computation, hence $O = \{2\}$.

## BOX 1
# Multi-client blind quantum computation

**Public information**
- A graph $G = (V, E, I, O)$ with input and output vertices $I$ and $O$, respectively.
- A partition $(I_j)_{j \in \{1,\ldots,n\}}$ of the input vertices, where $I_j$ belongs to client $j$.
- A partial order $\preceq$ on the set $V$ of vertices.

**Inputs**
- Client $j$ has a classical bit string $x_j \in \{0,1\}^{|I_j|}$.
- The $n$ clients collaboratively input a set of angles $(\phi_v)_{v \in V}$, and a flow $f$ on $G$ compatible with $\preceq$.
- The TTP and the server have no inputs.

**Protocol**
1. The $n$ clients send all of their inputs to the TTP.
2. The TTP and the server perform the BQC Supplementary Protocol 1. For every $|+_\theta\rangle$-state that the TTP would need to send to the server, they instead perform the following:
2a. The server prepares a $|+\rangle$-state.
2b. All parties perform one execution of the Collaborative Remote State Rotation Supplementary Protocol 2, where the server uses the $|+\rangle$-state as an input, and the TTP inputs the angle $\theta$.
2c. The server uses the output state in the BQC Supplementary Protocol 1.
3. The TTP distributes the classical output among the clients. The server sends the output qubits in $O$ to the designated clients, with the TTP providing the decryption keys.

## Security

In our protocol, the clients' quantum abilities are restricted to receiving single-qubit states, applying a random $z$-rotation to it, and forwarding it to the next client or server. From the perspective of an honest client, this behavior is exactly captured by the *Remote State Rotation* (RSR) functionality introduced in ref. [11]. The latter showed that RSR can indeed be used in the context of the BQC protocol to delegate a universal quantum computation with perfect blindness for input, output, and algorithm. This immediately implies the security of the proposed protocol with a single honest client and an untrusted server, as the instructions of the protocols exactly coincide. By a similar argument, security can be shown for the two-client and multi-client generalizations. In the spirit of the security proof in ref. [36], the sequential rotations of a single-qubit by all clients along the Qline can be seen as a collaborative version of RSR where the role of the RSR client is now taken by an entirely classical trusted virtual party, the TTP. The entire execution of the protocol then becomes one run of the BQC protocol between the TTP and the server. Finally, in real-world implementations, the TTP can be replaced by a classical SMPC protocol. Security follows by the composition of the above-mentioned building blocks which have all been proven to be composably secure in the Abstract Cryptography framework[50]. Further details about the security analysis of the full protocol, including a formal security proof, are provided in Supplementary Note 1. We stress that the source of quantum states is not required to be trusted. Indeed, we demonstrated that none of the parties involved can gain any information about the input, output, and computation details.

## Experimental apparatus

In Fig. 3, we describe the experimental apparatus we employ to implement the two-client protocol. A detailed time scheme of the protocol is reported in Supplementary Note 2 and in Supplementary Fig. 1, while a detailed description of the state preparation and measurement stages is discussed in Supplementary Note 3. A Sagnac-based source of polarization-entangled photons, i.e., server $S_1$, generates pairs of photons in the state defined in Eq. (1), where we encode the computational basis vector $|0\rangle$ in the photons' horizontal polarization ($|H\rangle$), and $|1\rangle$ in the vertical one ($|V\rangle$). The photon pairs are sent to the clients who apply their random rotations. The resulting state after these transformations is defined in Eq. (2). At each run of the protocol,

we set all random parameters through an ID Quantique quantum random number generator (QRNG). Both clients use liquid crystals (LCs) to apply their rotations, which are set in this preparation stage. The TTP is made up of a computer linked to a fast electronic circuit and stores all clients' parameters. With such information, the TTP precomputes the measurement angle $\delta_1$, and the first measurement station is set accordingly. Moreover, the TTP also pre-computes the two possible values for $\delta_2$, considering that the first outcome is still unknown, namely $\delta_2^\pm = \theta_2 + x_2\pi + r_2\pi \pm \phi_2$, to speed up the measurement step of the protocol. The two photons are sent to server $S_2$ of Fig. 3 where measurements of the form $M(\delta) = \cos(\delta)\sigma_x + \sin(\delta)\sigma_y$ are performed. The first one is made up of a quarter-wave plate (QWP), a half-wave plate (HWP), and a polarizing beam splitter (PBS). The two single-photon avalanche photodiodes (APD) of the first measurement station are connected to a fast electronic circuit that selects $\delta_2^+$ or $\delta_2^-$, according to the corrected outcome of the first measurement, $m_1^{\text{true}} = m_1 \oplus r_1$. In the second measurement station, we substitute the QWP with a Pockels cell (PC), i.e., a fast electro-optical modulator that performs the identity when no voltage is applied, while applying a phase shift between orthogonal polarizations when a voltage is applied. The second photon is delayed with respect to the first one by using a $\approx 65$ m single-mode fiber to enable feed-forward in the second measurement station. Finally, the second outcome is corrected according to $m_2^{\text{true}} = m_2 \oplus r_2$. Further details about the feed-forward system can be found in Supplementary Note 4 and in Supplementary Fig. 2. All events are collected through a coincidence box that records as two-fold coincidences all detector clicks occurring in a given time window.

## Experimental results

To show that the server cannot gain any information about the outcome of the computation, we suppose that the clients want to compute a given quantum function whose outcome is represented by the second qubit. We repeated the experiment for both qubits of the cluster state, but we show in the main text only the resulting density matrix for the second qubit. Details about the blindness of the first qubit can be found in Supplementary Note 5 and in Supplementary Fig. 3. We demonstrate blindness of the second qubit by keeping the measurement angle $\delta_1 = \pi$ fixed and by averaging over all density matrices resulting in the output qubit for different initial rotation

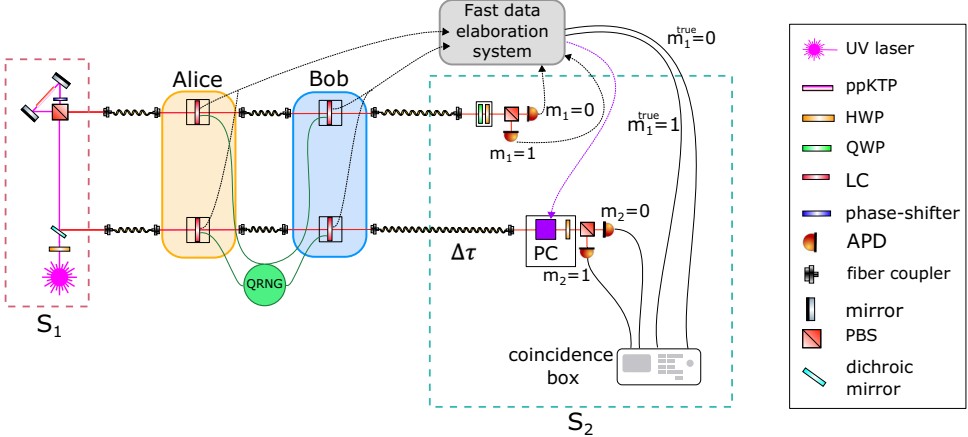

**Fig. 3 | Experimental apparatus.** The state defined in Eq. (1) is generated through a Sagnac-based source of entangled photons, where horizontal polarization ($|H\rangle$) encodes the state $|0\rangle$ while vertical polarization ($|V\rangle$) encodes the state $|1\rangle$. The photons are first sent to Alice and Bob who perform single-qubit rotations by means of liquid crystals (LC). To make the clients' choices random, the clients' secret parameters are chosen by means of a quantum random number generator (QRNG). Then, the two photons are sent to Server 2 ($S_2$). While the second photon is delayed through $a \approx 65$ m-long single-mode fiber, the first photon is measured, by using a sequence of a quarter-wave plate (QWP), a half-wave plate (HWP), and a polarizing beam splitter (PBS). The second measurement station, instead, is composed of a Pockels cell (PC), a HWP, and a PBS. A fast data elaboration system, constituted by a computer and a fast electronic circuit, embodies the TTP. The two detectors of the first measurement station are linked to such a system, that directly activates the PC with a suitable high-voltage (HV) pulse according to the desired second measurement basis. All outcomes are collected through a coincidence box that records as two-fold coincidences all events occurring in a given time window.

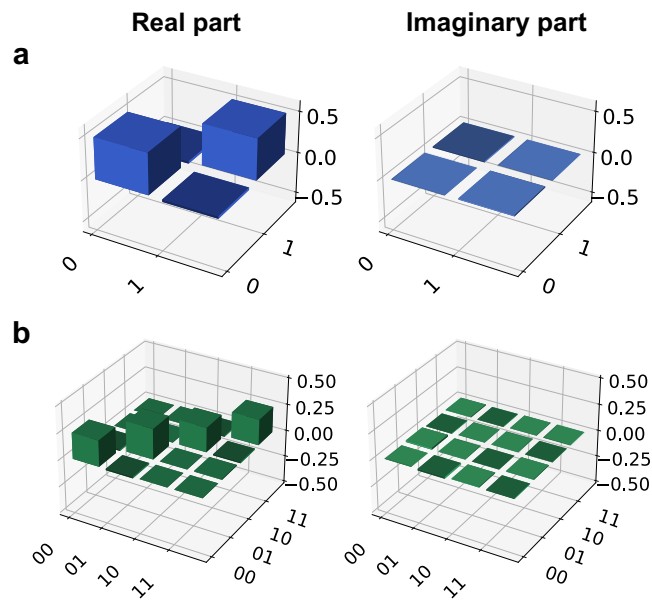

**Fig. 4 | Demonstration of blindness. a** Density matrix of the second qubit averaged over all possible $\theta_1^A$ and $\theta_1^B$ configurations. **b** Density matrix of the two-qubit initial state, averaged over all possible values of $\theta_1^A$ and $\theta_2^B$.

angles $\theta_1^j$, where $j =$ A, B, namely 64 combinations. The density matrix in Fig. 4a shows the resulting quantum state, which has a fidelity with a single-qubit completely mixed state amounting to $F_2 = 0.99870 \pm 0.00003$, while the measured Von Neumann entropy is $S_2 = 0.9963 \pm 0.0001$, to be compared with the expected value of 1 for a completely mixed single-qubit state. Furthermore, we demonstrate the blindness of the whole initial two-qubit cluster state, by averaging over all density matrices corresponding to 64 combinations of the initial z-rotations, for parameters $\theta_1^A$ and $\theta_2^B$, while keeping $\theta_1^B, \theta_2^A = 0$. We stress that these combinations are enough to demonstrate blindness, as the random rotations on each qubit still take all possible values in the set $\mathcal{A}$. For the two-qubit state, whose density matrix is shown in

Fig. 4b, we estimated a fidelity $F = 0.99433 \pm 0.00003$ with the completely mixed state and with a Von Neumann entropy of $S = 1.9836 \pm 0.0001$, to be compared with the expected value of 2 for a completely mixed two-qubit state. All density matrices are retrieved from raw experimental data through quantum state tomography[51]. Further details about the blindness of the second qubit and the full initial state are reported in Supplementary Notes 6 and 7 and in Supplementary Figs. 4 and 5.

Let us now consider the scenario where the clients want to compute a quantum function. In this case, we take the second qubit as the outcome of the computation, by preparing it in the state $|+\rangle$. We perform the computation $\phi_1 = \pi/4$, with input bits $x_1, x_2$ set to 0. We set the clients' secret parameters as $\theta_1^A = \pi/2$, $\theta_1^B = \pi/4$ and $r_1^A = r_1^B = 0$. We show our results in Fig. 5. The estimated fidelity with the ideal state amounts to $F_{\pi/4} = 0.972 \pm 0.003$. In Supplementary Notes 5 and 6, we show instances of single-qubit quantum state tomographies of the two qubits and their fidelity with respect to the theoretical expectations. We then demonstrate the correctness of the computation for classical functions. The algorithm performed over input data $x_1, x_2$ is characterized by the two true measurement angles $\phi_1, \phi_2$. Choosing $x_1$ or $x_2$ equal to 1 has simply the effect of inverting the minima and the maxima of the distributions. In Fig. 6, we show ten different probability distributions obtained by trying ten different combinations of the algorithm parameters $(\phi_1, \phi_2, x_1, x_2)$, and the comparison with the ideal and the noisy model case. All the obtained results are in good agreement with our noisy model and follow qualitatively the ideal expectations. Small deviations from the expected values are mainly due to the visibility of the quantum state at the end of the Qline and to imperfections coming from the non-ideal electro-optical modulators employed, i.e., the LCs and the PC (further details about our noisy model and the protocol correctness validation are shown in Supplementary Notes 8 and 9 and in Supplementary Figs. 6 and 7).

## Discussion

In this work, we proposed a multi-client version of the BQC protocol[7] and experimentally demonstrated it in a two-client setting. We first simplified the protocol described in refs. 36,40 to tailor it to the photonic Qline network introduced in ref. 45. To this end, we studied a

photonic platform equipped with a source of polarization-entangled photon pairs, an active feed-forward system, and a fiber-based structure to connect the involved parties. In our scheme, the clients only need to apply single-qubit rotations. Within this setup, we computed the outcomes of ten different classical functions, by changing the input data and the algorithm, and compared the results with a noisy model compatible with our experimental conditions. Also, we demonstrated the correctness of the protocol when the function to be computed has a quantum output. Finally, we showed that the server cannot gain any information about the inputs of the clients or the outcome of the computation.

Our proof-of-concept demonstration can represent a step forward toward the realization of a scalable and secure quantum cloud access infrastructure with multiple clients. Indeed, in a real-world

protocol, the necessary classical communication as part of the SMPC in between the measurements would considerably increase the time latency, in particular, if run over a slow network, such as the internet. Therefore, to make the experimental realization of the proposed protocol feasible, we replaced the SMPC with a fast electronic data elaboration circuit to both reduce the communication rounds and compensate for the absence of quantum memory. This allowed us to reduce the delay between measurements to a manageable level. Scaling up the size of the computation would require more qubits and their communication from the photon source past all clients to the server. In principle, this could be realized in two ways that can eventually be combined. First, the qubits could be sent all at once, which would require the clients to be able to apply rotations to multiple states at once. Alternatively, the qubits could be sent sequentially, one at a time. However, this second option would require the ability to store or delay them until the clients were able to adjust their rotation gates since every qubit is rotated by a different angle. In our demonstration, we opted for the first option as it represented the optimal way to minimize time latency and, consequently, photon loss. On the other hand, from the protocol point of view, adding clients only requires adding rotation stations along each Qline, and our general proof technique covers the security aspects of such extensions. On the experimental side, instead, in our scheme, this would only imply handling more optical losses, which can be overcome with a brighter quantum source, and an accurate characterization of the optical elements that perform the single-qubit rotations. This would not imply any substantial change even in the design of the fast data elaboration circuit, as we show in Supplementary Note 4. Moreover, the choice of adopting an optimized feed-forward system for measurement adaptivity in our setup is not only crucial to ensure blind and deterministic computation, but also to scale up the protocol. Indeed, post-selection schemes would require an exponentially growing number of measurements depending on the dimension of the quantum system, which would affect significantly the possibility of applications to larger quantum systems. While our implementation guarantees the privacy of the inputs provided by the clients, the outcome of the computation is not verified. We leave the addition of verification to the proposed protocol as future work. One possible path towards verification with Qline architecture might be the employment of the recently introduced *dummyless* testing technique from ref. 36. However, as of now, the question of whether states prepared by rotation-only clients are sufficient for verification remains

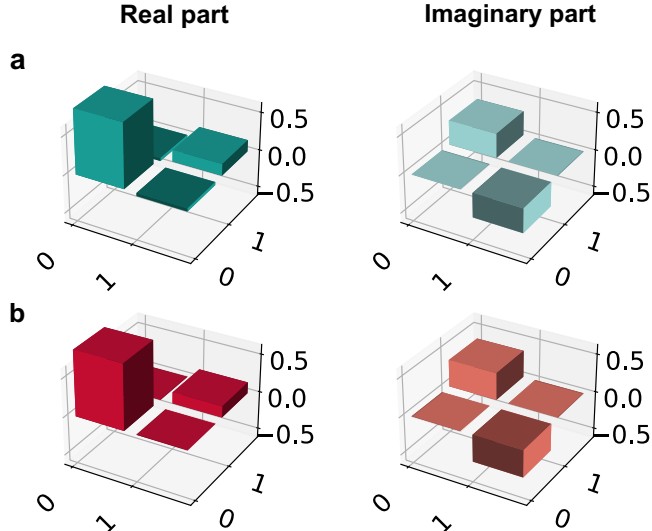

**Fig. 5 | Computation of a quantum function.** In this figure, we show the results of the computation of a quantum function. Bob chose $\phi_1 = \pi/4$, while the clients' random parameters are $\theta_1^A = \pi/2$ and $\theta_1^B = \pi/4$. Both input bits $x_1, x_2$ are set to 0. The first qubit is thus measured in the basis $\delta_1 = \pi$. The experimental density matrix is shown in panel **a**, while the theoretical one is shown in panel **b**.

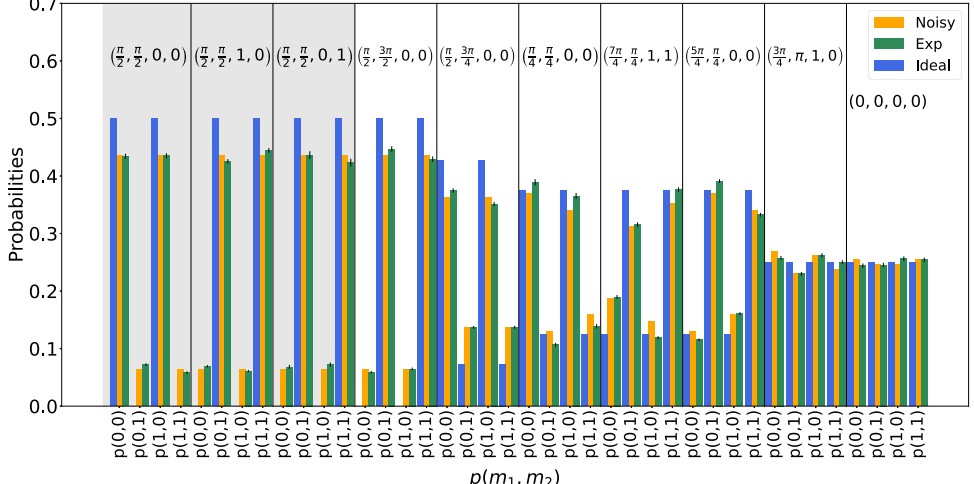

**Fig. 6 | Computation of a classical function.** In this bar plot, we show ten different measurement angles and Alice's input combinations $(\phi_1, \phi_2, x_1, x_2)$. In the gray region, we kept the algorithm fixed while changing the input data, to show the changes in both the expected and experimental distributions. In the white region, instead, we changed both algorithms and input data. The uncertainties on the experimental frequencies were obtained assuming Poissonian statistics, and the black bars correspond to one standard deviation. The eventual absence of black bars means that the uncertainty was too small to be visible in this plot.

open, as ref. 11 only showed that these kinds of states are enough to achieve blindness.

We believe that this work has insightful implications both from a theoretical and an experimental point of view. From a theoretical perspective, it constitutes a strong encouragement toward the development of collaborative computational algorithms over distributed quantum networks, as well as investigations about their verification. From an experimental point of view, instead, it represents a step forward toward the applications of photonic linear quantum networks as building blocks for more complex networks, toward the realization of a large and densely connected quantum cloud.

## Methods

Photon pairs are generated in a parametric down-conversion source, composed of a 25-mm-thick periodically poled Potassium Titanyl Phosphate (ppKTP) crystal inside a Sagnac interferometer. The source uses a Toptica continuous-wave diode laser with a wavelength equal to 405nm. Both photons are generated at a wavelength equal to 810nm. To test the quality of the bipartite state generated by $S_1$, we perform a CHSH Bell test[52] and obtain a Bell parameter equal to $2.752 \pm 0.006$. The generated photons are filtered in wavelength and spatial mode by using, respectively, narrow-band filters and single-mode fibers. The PC is a $LiNbO_3$ crystal made by the Shangai Institute of Ceramics having a rise-time equal to 90 ns. A fast electronic circuit transforms signals coming from the detectors of the first measurement station into high-voltage calibrated pulses, needed to activate the PC. The amount of delay on the second photon was evaluated considering the response time of the detectors, the speed of the signal transmission through a single-mode fiber, whose refraction index is $\approx 1.45$, and the activation time of the PC. Therefore, we used a $\approx 65\,m$ long single-mode fiber that allows a delay of $\approx 320\,ns$ of the second photon with respect to the first. The voltages applied to the PC to insert a phase shift equal to $\pi/4, \pi/2, 3\pi/4$ were, respectively, $V_{\pi/4} = 650V, V_{\pi/2} = 850V, V_{3\pi/4} = 1100\,V$. Further details about the feed-forward system are given in Supplementary Note 4. Our experiment is performed shot-by-shot, namely, each event of our data takings is characterized by a different randomly chosen set of initial parameters $\theta_i^j, r_i^j$, for $i = 1, 2$ and $j = A, B$, while the algorithm ($\phi_i, x_i$, for $i = 1, 2$) is kept fixed for each data taking.

## Data availability

The authors declare that the data supporting the findings of this study are available upon request.

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

## Acknowledgements
All authors acknowledge the support of the European Union's Horizon 2020 research and innovation program through the FET project PHO-QUSING ("PHOtonic Quantum SamplING machine"—Grant Agreement No. 899544). D.L. and E.K. acknowledge the support of the ANR research grant ANR-21-CE47-0014 (SecNISQ).

## Author contributions
B.P., D.L., L.L., G.C., N.S., M.K., F.S., and E.K. contributed to developing the ideas and discussions explored in the paper. B.P., D.L., M.K., and E.K. contributed to the design of the protocols. B.P., L.L., G.C., G.M., N.S., and F.S. contributed to the experimental implementation. All authors contributed equally to the writing, editing, and final revision of the paper.

## Competing interests
E.K. and M.K. are founding members of VeriQloud SAS. The remaining authors declare no other competing interests.
