## [Peer Review File · Nature Communications]

Multi-client distributed blind quantum computation with the Qline architectureREVIEWER COMMENTS

Reviewer #1 (Remarks to the Author):

Blind quantum computation (BQC) allows a client to perform secret computation tasks on a remote quantum computing server. There have been several works on proposing protocols, investigating the minimal resource requirements for client, reducing the computation resource cost, etc. This work focuses on a different while important aspect: the case of multi-client. The authors adopted the Qline configuration to propose the multi-client BQC protocols and then demonstrate the two-client case using optical platforms. The manuscript is well-written with a clear introduction to theoretical protocols. The results are solid and should be considered for high-level journals such as Nature Communications.

Before acceptance, I hope the authors can address the following questions and comments.

1. Is the source trusted or not?
2. On the theoretical side, the orchestrator is not shown in figures. It is hard to imagine the role of orchestrator in the protocol.
3. The authors noted that “the orchestrator from Protocol 1 must be trusted” after introducing Protocol 1. Can this orchestrator be one of the clients? How would an untrusted orchestrator break down the security of the BQC protocol?
4. Following the second question, the authors provided a method to remove the trust assumption on the orchestrator by involving classical SMPC protocol. I wonder is there any difficulty when one really does this.
5. In the scheme all the clients use the same source and encode the generated state sequentially. I wonder if the protocol can be realized with independent sources.
6. In the experiment a two-client protocol is demonstrated. Is there any technical problem if carrying out a more-than-three-client protocol?

Reviewer #2 (Remarks to the Author):

This paper attempted to propose and experimental demonstrate a lightweight multi-client blind quantum computation (BQC) protocol based on a linear quantum network configuration (Qline). It claimed to satisfy scalability, low-loss, and compatibility

simultaneously. The work is interesting and significant, but some problems should be addressed.

1. The proposed multi-client BQC protocol seems to be mainly based on the Qline quantum network link configuration in Ref. [43] which has not published yet. But the knowledge about Qline has less introduction and it is better to introduce it in more detail.

2. The work combines some ideas in difference references and should give clearly and correctly where such methods or idea comes from and cite them. For example, the manuscript stated that it presents a tailor-made multi-party BQC protocols such that the clients in the Qline only own trusted single qubit rotation devices and though the idea comes from the Qline network architecture. But, as far as we know, this idea has already been proposed in other researchers' work entitled as "Blind quantum computation where a user only performs single-qubit gates (Optics and Laser Technology 2021)" where indeed an untrusted server generated some quantum states and sent to any client who can only apply arbitrary single-qubit operations on the incoming qubits. The differences and similarities between them should be clarified.

3. In Part A, Section II, Alice's inputs are x_1 and x_2 , while Bob's inputs are ϕ_1 and ϕ_2 . Is it also right the other way round? The range of values for these parameters also should be given. In addition, in the two-client protocol, Bob has the inputs ϕ_1 and ϕ_2 , but in Protocol 1, the client j has the inputs $x_j \in \{0,1\}^{(|I_j|)}$. Is there anything wrong? I think they should be given more clearly.

4. Several participants are involved in the proposed multi-client BQC protocol, such as some clients, Server 1, Server 2, TTP, and Orchestrator. What roles they played and whether they are honest or not is confusing.

5. Generally, each client's dataset size is different in a federated machine learning task. But in Protocol 1, the input of each user should be the same size. So, is it still be suitable for dealing with federated machine learning tasks?

6. The security analysis should be more clear and complete. For example, a BQC protocol is said to satisfy blindness if all the data of users including the input, output and the algorithm can be kept private. But the current edition seems to only show the server cannot gain any information about the input and the outcome of the computation.

Reviewer #3 (Remarks to the Author):

The paper proposes a novel lightweight multi-client protocol for blind quantum computation based on a linear quantum network configuration (denoted as Qline). Moreover, the paper presents a two-client example that was implemented experimentally.

The work is original and will be of significance to the field of quantum networking and blind quantum computing. With respect to the established literature, the work has two relevant features: the clients only own trusted single-qubit rotation devices, and have to perform just a new layer of encryption to the flying qubits that traverse the Qline.

The conclusions and claims are supported by analytical proofs and - for the two-qubit example - by experimental data that are particularly impressive.

The proposed methodology is sound, from both the analytical and experimental points of view.

The organisation of the manuscript is good, in general, but may be further improved.

1) Fig. 1 (inspired by Fig. 5 in [48]) requires reading [48] to be fully understood. In particular, the Server part in the figure is not clearly describe by the caption. Also, n is used to indicate the number of qubits that the client sends to the server, but later in the manuscript n is the number of clients. I would suggest to use m to indicate the number of qubits that the client sends to the server.

2) The two-client protocol described in section II.A does not appear to be a 2-client instance of the multi-client protocol described in section II.B. More precisely, in the two-client protocol there is one source of entangled pairs that feeds Alice. In the multi-client scheme, there are m sources, each one distributing single qubits. This mismatch should be clarified,

otherwise it seems misleading to state that the multi-client protocol is a "generalisation" of the two-client protocol. In my opinion, it would be better to start from the most general case, then illustrate the two-qubit example (which is the one that was experimentally implemented).

3) In the introduction, it is stated that "the quantum resource is first generated by a potentially untrusted server, then distributed to the clients". However, in the manuscript, the potential unreliability of the quantum sources is not discussed. Actually, it seems that the quantum sources are always trusted, like the classical orchestrator. But while the orchestrator could be replaced by a classical SMPC protocol, it seems there is no way to replace the trusted sources. The authors should discuss this potential issue.

4) The discussion about the scalability of the protocol could be improved by describing how the approach based on a "fast electronic data elaboration circuit" (described in Supplementary Material S4) would scale with more than 2 clients.

The experimental methods are presented in a detailed fashion (the Supplementary Materials are quite rich), allowing for the reproduction of the work.

Reply to Reviewer 1

Blind quantum computation (BQC) allows a client to perform secret computation tasks on a remote quantum computing server. There have been several works on proposing protocols, investigating the minimal resource requirements for client, reducing the computation resource cost, etc. This work focuses on a different while important aspect: the case of multi-client. The authors adopted the Qline configuration to propose the multi-client BQC protocols and then demonstrate the two-client case using optical platforms. The manuscript is well-written with a clear introduction to theoretical protocols. The results are solid and should be considered for high-level journals such as Nature Communications. Before acceptance, I hope the authors can address the following questions and comments.

We thank the Reviewer for their careful reading of our work and for acknowledging its interest and solidity. We will now respond point-by-point to their comments and questions.

1. Is the source trusted or not?

In our approach, the source is not required to be trusted, and we thank the Reviewer for pointing out this crucial strength of our protocol, which is now better explained in the main text. Let us also elaborate on this further here.

The protocol we designed ensures that blindness is satisfied against not only the server but also potentially dishonest clients that collude with it, even if the source of states is not trusted. Indeed, from the blindness perspective, we demonstrate that the server providing quantum states, just like any other party involved in the protocol except the orchestrator, cannot get any information about input, output, and computation details. To clarify this point, we made the following changes to the main text:

- In the “State preparation” paragraph of subsection IIA, we added the adjective ***untrusted*** when describing the source of maximally entangled states.
- We added the following sentence at the end of subsection IIC:
“We stress that the source of quantum states is not required to be trusted. Indeed, we demonstrated that none of the parties involved can gain any information about the input, output, and computation details.”

2. On the theoretical side, the orchestrator is not shown in figures. It is hard to imagine the role of orchestrator in the protocol.

In the protocols we introduce, the orchestrator is a trusted third party which could in principle be realized in several and different experimental ways. Its role is to secure the classical communication between the clients and the server, by receiving all the clients’ secret parameters and computing the measurement angles to be performed by the server at each round. In this way, the honest parties can avoid trusting any potentially dishonest client and/or server, since only the orchestrator needs to be trusted. Accordingly, in our manuscript, we use the terms “orchestrator” and “trusted third party” to indicate the same conceptual role. However, to make this role clearer, we did the following changes in the main text:

- We now uniformed the two definitions by using everywhere “trusted third party (TTP)” instead of “orchestrator”.
- we rephrased a sentence in the introduction of section II as highlighted below:
“Therefore, to optimize the storage time, in our implementation, we substitute classical SMPC with a trusted third party (TTP) acting as an orchestrator and securing the communication between the clients and the server reducing the number of rounds, while the blindness of the protocol is still proven against any strict subset of colluding malicious adversaries.”
- we changed a sentence in subsection IIB as highlighted below:
“As all communication from this point forward is entirely classical, the TTP orchestrates the remainder of the protocol by governing the instructions to the server.”

3. The authors noted that “the orchestrator from Protocol 1 must be trusted” after introducing Protocol 1. Can this orchestrator be one of the clients? How would an untrusted orchestrator break down the security of the BQC protocol?

The orchestrator is a purely classical party (different from the clients in our protocol, which also have limited quantum capabilities), in charge of securing classical communication between clients and server, and ensuring that no data is leaked to potentially malicious participants. In theory, it is implemented through classical SMPC and, hence, it is not trusted, as SMPC provides the required trust assumption. However, in our implementation, integrating a full classical SMPC protocol to replace the task of orchestration would lead to severe constraints on the quantum communication timeline. Indeed, it would require additional rounds of classical communication among the clients, which would translate into longer storage time for the qubits. For these reasons, for this proof-of-concept, we are assuming it to be trusted as a placeholder for alternative implementations based on either classical SMPC or hardware tokens. To make this point clearer, we made the following changes to the main text:

- We changed the following sentence in subsection IIB as highlighted in bold below:

“We also point out that the orchestration of Protocol 1 must be performed by an entirely classical trusted third party that secures the secret parameters input by the clients as well as the measurement outcomes, which should not be leaked.”

4. Following the second question, the authors provided a method to remove the trust assumption on the orchestrator by involving classical SMPC protocol. I wonder is there any difficulty when one really does this.

As demonstrated in [1] (now [35] in the main text), SMPC represents a powerful tool to allow for multi-client BQC protocols such that the full computation details are blind to the server as well as to potentially dishonest clients that collude with it. However, practically speaking, the implementation of such a classical functionality for controlling the quantum devices in our platform would require additional rounds of classical communication among the quantum clients, which would translate into longer latency times between each round of the BQC protocol. Hence, from an experimental point of view, implementing SMPC in between each round of the protocol would enlarge the time during which qubits must be coherently stored, which represents a major technical challenge. For this reason, we proposed a modification of the protocol in [1], allowing for a trade-off between the trust assumptions needed to ensure security and the latency between the protocol rounds. Indeed, we substituted SMPC among the clients with a single classical trusted third party, while keeping the same modular architecture in place for future implementations with quantum memories, that could enable a full SMPC replacement for this trusted classical module with no further difficulty.

- To emphasize this concept, we added the following sentence in section II:

“Our solution represents a trade-off between hardware trust assumptions and time latency between rounds. However, the trust assumption on the TTP can be dropped without modifying our scheme, by simply replacing it with classical SMPC, if we run the computation on a quantum server equipped with quantum memory.”

5. In the scheme all the clients use the same source and encode the generated state sequentially. I wonder if the protocol can be realized with independent sources.

Actually, one of the strengths of the Qline network resides in its modularity. Indeed, it is possible to integrate it into various kinds of distributed architectures, also involving different and independent quantum state sources. In detail, in Fig. 2c of the main text, as well as in the description given in section “B. Generalization to the multi-client scenario”, we present such a scalable architecture with extended protocols and, more importantly, a full security proof that covers all such extensions including the special case raised by the Reviewer, while covering more general cases. Such extended protocols and security proofs can be tailored to different platforms. Indeed, assuming that the server has full quantum capabilities, it can entangle them into a suitable resource state and perform the computation.

- To make this possibility clearer, we added the following sentence in section IIB:

“Note that, here, we treat a more general case with respect to the previous one, hence the security proof we provide, which assumes a fully quantum server and independent and arbitrarily distant qubit sources, also holds for the implemented protocol.”

6. In the experiment a two-client protocol is demonstrated. Is there any technical problem if carrying out a more-than-three-client protocol?

We thank the Reviewer for pointing out a relevant aspect regarding the scalability of our protocol. Even if, for our proof-of-concept, we implemented the two-client setting, the only technical challenge when inserting more clients in the Qlines is related to the coupling losses at each client station, but this shortcoming can be overcome by using brighter quantum sources, or fibered rotation stations. Moreover, as the number of clients increases, the number of optical elements to perform the rotations increases as well, thus requiring accurate control over each of them to properly perform the required (and repeatable) measurements. From the protocol point of view, there is no extra difficulty, since adding clients requires adding qubit rotation stations along each line, as suggested by Fig. 2b of the main text. This change is also easily handled by the fast data elaboration circuit that embodies the TTP. Indeed, as described in Supplementary Material section S4, the number of parameters entering and exiting the circuit would not change if the number of clients increases. Furthermore, our general proof technique presented in subsection IIC already covers the security aspects of such extensions. To emphasize the possibility to scale up our protocol in the main text, we modified it as follows:

- We added the following sentence at the end of the “Scalability” paragraph in the “Discussion” section (section V):

“On the other hand, from the protocol point of view, adding clients only requires adding rotation stations along each Qline, and our general proof technique covers the security aspects of such extensions. On the experimental side, instead, in our scheme, this would only imply handling more optical losses, which can be overcome with a brighter quantum source, and an accurate characterization of the optical elements that perform the single-qubit rotations. This would not imply any substantial change even in the design of the fast data elaboration circuit, as we show in Supplementary Information section S4.”

- Furthermore, we added the following sentence in section S4 of the Supplementary Information:

“We now demonstrate that such a circuit would not need to be changed in its structure when having n clients. In our protocol, to ensure the privacy of all clients, each of them needs to rotate all the qubits involved in the computation. Then, one would first have to compute the cumulative secret rotation angle θ_i for the i -th qubit as $\theta_i = \sum_{j=1, \dots, n} \theta_i^{C_j}$ and the cumulative random bit $r_i = \bigoplus_{j=1, \dots, n} r_i^{C_j}$, where j indicates the j -th client. The classical input data x_2 and algorithm details ϕ_2 can be summed up analogously depending on the subset of clients that provided such information. Such cumulative parameters can then be placed in the coefficients A and B for the computation of the blind measurement angle δ_2 . The remainder of the computation would not change.”

Reply to Reviewer 2

This paper attempted to propose and experimental demonstrate a lightweight multi-client blind quantum computation (BQC) protocol based on a linear quantum network configuration (Qline). It claimed to satisfy scalability, low-loss, and compatibility simultaneously. The work is interesting and significant, but some problems should be addressed.

We thank the Reviewer for their careful reading of our manuscript and for acknowledging the interest and significance of our work. We will now address point-by-point their comments and questions.

1. The proposed muti-client BQC protocol seems to be mainly based on the Qline quantum network link configuration in Ref. [43] which has not published yet. But the knowledge about Qline has less introduction and it is better to introduce it in more detail.

In the following, we would like to explain the motivation behind the current introduction of the Qline architecture. Our protocol shares some defining features of its network topology with the protocol proposed in [2] (now ref. [44] in the main text), which is why we are calling its architecture Qline. These features are (i) the linear configuration of the clients, and (ii) the clients' restriction to perform single-qubit operations. As a consequence, our protocol inherits several of its practical advantages, such as versatility and scalability. Despite these commonalities, our protocol significantly differs from previous works, including [2]. In particular, it constructs a functionality that is fundamentally different from QKD, for which the Qline paradigm was used previously. For this reason, our manuscript provides a full security proof that does not depend on the mentioned reference. Given the differences to the previous proposal, we believe that details of the Qline protocol in [2] beyond the basic network topology are not relevant for this work and therefore we decided to omit them from the manuscript.

2. The work combines some ideas in difference references and should give clearly and correctly where such methods or idea comes from and cite them. For example, the manuscript stated that it presents a talilor-made multi-party BQC protocols such that the clients in the Qline only own trusted single qubit rotation devices and though the idea comes from the Qline network architecture. But, as far as we know, this idea has already been proposed in other researchers' work entitled as "Blind quantum computation where a user only performs single-qubit gates (Optics and Laser Technology 2021)" where indeed an untrusted server generated some quantum states and sent to any client who can only apply arbitrary single-qubit operations on the incoming qubits. The differences and similarities between them should be clarified.

We acknowledge that the reference mentioned by the Reviewer is related to our work and we now cite it in the revised version of the manuscript. However, it is important to emphasize that we do not use any of the previously presented protocols, such as UBQC with one client [3] (now [7] in the main text), the QLine for QKD [2] or previously presented quantum SMPC-based (QSMPC) protocols involving several clients [1, 4, 7] (now [35-39] in the main text) as subroutines in our work. Indeed, on one hand, we designed a new multi-client QSMPC protocol inspired by elements appearing in previous literature (such as the underlying module with rotating clients [8] (now [11] in the main text), the Qline architecture [2, 9] (now [44-45] in the main text), and others) that we mention as sources of inspiration. But, on the other hand, our work is developed from scratch and independent, and, as a result, we needed to provide a self-contained security proof, which we did. Indeed, even if the protocols appearing in the papers we mention are proven in the composable security framework, the security proof and the properties of our protocol cannot be directly derived from them, as the latter is not a mere composition of such previous works.

- Following the Reviewer's suggestion, we added the above explanation to the introduction:

"The model for BQC in which the client only performs single-qubit gates was already proposed in [11,12]. However, although our protocol is inspired by the aforementioned works, we emphasize that its security properties cannot be directly derived from those BQC protocols as our multi-client setting is not a mere composition of such previous works. Therefore, in this work, we also provide a full security proof, as we do in Section IIC."

3. In Part A, Section II, Alice's inputs are x_1 and x_2 , while Bob's inputs are ϕ_1 and ϕ_2 . Is it also right the other way round? The range of values for these parameters also should be given. In addition,

in the two-client protocol, Bob has the inputs ϕ_1 and ϕ_2 , but in Protocol 1, the client j has the inputs $x_j \in \{0, 1\}^{(|I_j|)}$. Is there anything wrong? I think they should be given more clearly.

We thank the Reviewer for their suggestions. In our implementation, x_1 and x_2 are two classical bits of information, while ϕ_1 and ϕ_2 are angles chosen from the set $\mathcal{A} = \{0, \pi/4, 2\pi/4, \dots, 7\pi/4\}$. We added this clarification in the main text as follows:

- in the subsection IIA, we modified the following sentence as highlighted below in bold:
 “Consider Alice and Bob wish to run a joint computation on a remote server. Alice has **two private classical bits of information**, x_1 and x_2 , while Bob has private gate parameters ϕ_1 and ϕ_2 , **chosen from the set $\mathcal{A} = \{0, \pi/4, 2\pi/4, \dots, 7\pi/4\}$** , and the target joint circuit is [...]”
- We deleted a repeated definition of the set \mathcal{A} in the same subsection.

Furthermore, the distribution of the input data can change as well as the input data size. Indeed, the choice we made in our manuscript is only an example and can be easily generalized. For instance, for the first qubit, Alice and Bob could both insert respectively the two bits of classical information x_1^A and x_1^B , encoding them through a total single-qubit transformation described by the operator $Z^{x_1^A \oplus x_1^B}$. Analogously, they can respectively provide two measurement angles ϕ_1^A and ϕ_1^B , thus providing the total measurement angle $\phi_1 = \phi_1^A + \phi_1^B$, which still belongs to the set \mathcal{A} . Similar reasonings apply to qubit 2.

In the first version of the manuscript, such a concept is written in subsection IIB, where we state that: 1) the set of measurement angles ϕ_v can be publicly known or jointly input by any subset of clients and that, in the latter case, such measurement angles can be kept hidden, and 2) that the clients’ input is the bit string $x_j \in \{0, 1\}^{(|I_j|)}$. To make the link between the two-client and the n -client protocols clearer, in this revised version, we made the following change:

- we added the following sentence to subsection IIA :
 “**Indeed, the distribution and the size of the input data are flexible, making this protocol suitable even for federated machine learning tasks. For example, each client C_j could provide one measurement angle $\phi_i^{C_j}$ for each qubit i , and one classical bit $x_i^{C_j}$. In this case, the cumulative private measurement angle applied to the i -th qubit would be $\phi_i = \sum_j \phi_i^{C_j}$, still in the set \mathcal{A} , while the initial encoding of classical data will be described by the operator $Z^{\bigoplus_j x_i^{C_j}}$. Also, not all clients are required to provide input data for each qubit.**”

4. Several participants are involved in the proposed multi-client BQC protocol, such as some clients, Server 1, Server 2, TTP, and Orchestrator. What roles they played and whether they are honest or not is confusing.

In our protocol, the TTP and the Orchestrator embody the same role, since the TTP is a trusted third party that orchestrates the full protocol by securing the classical communication between the clients and the server, thus allowing one to drop any trust assumption on all the other parties involved. To avoid any misunderstandings, we now uniformed the two definitions by using everywhere “trusted third party (TTP)” instead of “orchestrator”. However, we need to emphasize that the TTP has the role of an orchestrator for our multi-client BQC protocol. Therefore, we made the following changes to the main text:

- we rephrased a sentence in subsection IIA as highlighted below:
 “Therefore, to optimize the storage time, in our implementation, we substitute classical SMPC with a trusted third party (TTP) that **acts as an orchestrator by securing** the communication between the clients and the server reducing the number of rounds, while the blindness of the protocol is still proven against any strict subset of colluding malicious adversaries.”
- we changed a sentence in subsection IIB as highlighted below:
 “As all communication from this point forward is entirely classical, the TTP **orchestrates the remainder of the protocol by governing** the instructions to the server.” “

5. Generally, each client’s dataset size is different in a federated machine learning task. But in Protocol 1, the input of each user should be the same size. So, is it still be suitable for dealing with federated machine learning tasks?

In our protocol, each client has to contribute with their own secret parameter to mask all the qubits. Indeed, due to the global correlation between all qubits and the flow of information in such BQC protocols, even if a particular qubit of the computation is not directly linked to a given input of a given client, independent masking of each qubit is necessary to ensure that no malicious party can collude to derive any information from such partial encoding. Hence, from this point of view, the masking keys are generic and have the same size for each client, i.e. the size of the underlying resource state for the computation. However, encoding input data in all qubits does not bring any improvement from the security point of view, hence, as long as it is not needed for the sake of a correct computation, it is not at all necessary. On the contrary, the input data can be distributed in different ways among the clients, and not all clients need to provide data for each qubit, as already discussed in point 3. Therefore, from this point of view, we believe that our protocol is still suitable for federated machine-learning tasks.

- We included this discussion in the sentence added to the main text to reply to point 3, where we state that:
“Indeed, the distribution and the size of the input data are flexible, making this protocol suitable even for federated machine learning tasks. [...]”

6. The security analysis should be more clear and complete. For example, a BQC protocol is said to satisfy blindness if all the data of users including the input, output and the algorithm can be kept private. But the current edition seems to only show the server cannot gain any information about the input and the outcome of the computation.

Our protocol does also satisfy blindness for the algorithm, on top of hiding the inputs and outputs of the computation. As is common in the study of secure delegated quantum computation (see e.g. [10]), the (classical) description of the algorithm can be considered another input to a universal computation, which accepts as inputs both the input to the target algorithm as well as a classical description of the target algorithm itself, and returns the output of the target algorithm when applied to the specified input. When using a universal resource state in the protocol, such as brickwork states, the algorithm is indeed entirely hidden and the only information which is leaked to the server is an upper bound on the size of the computation.

As an additional advantage of the modularity of our protocol, it is not actually necessary to use universal resource states when willing to compromise on the blindness of the algorithm. In this way, clients could opt to use cheaper, non-universal resource states that are closer to the actual requirements of the target algorithm rather than using more expensive universal resource states. This however will leak some additional information to the server as it restricts the class of algorithms that could potentially be performed using the chosen resource state.

We would like to remark that our protocol is optimal in the sense that both of the aforementioned restrictions are necessary. First, any non-trivial protocol that delegates quantum computations must leak an upper bound on the size of the performed computation as the server naturally knows the expended resources on its side. Secondly, for similar reasons, any protocol that blindly delegates quantum computations and that does not leak any additional information about the target algorithm must require universal resources, as otherwise the server could exclude some algorithms based on the knowledge that they could not have been performed using only the provided, non-universal resource. To make this clearer, we made the following changes to the manuscript:

- We modified a sentence in section IIC of the main text as shown below:
“The latter showed that RSR can indeed be used in the context of the BQC protocol to delegate a universal quantum computation with perfect blindness for input, output, and algorithm.”
- We modified the following in subsection 1C of the Supplementary Information as highlighted below:
“Protocol S4 is a version of this protocol that does not necessarily require universal resource states, but rather works for any graph that is usable to implement the target computation. In this way, clients can opt to use cheaper, non-universal resource states that are closer to the actual requirements of the target algorithm rather than using more expensive universal resource states. This however will leak some additional information to the server as it restricts the class of algorithms that could potentially be performed using the chosen resource state. Of course, it remains possible to use universal resource states for full blindness.”

Reply to Reviewer 3

The paper proposes a novel lightweight multi-client protocol for blind quantum computation based on a linear quantum network configuration (denoted as Qline). Moreover, the paper presents a two-client example that was implemented experimentally. The work is original and will be of significance to the field of quantum networking and blind quantum computing. With respect to the established literature, the work has two relevant features: the clients only own trusted single-qubit rotation devices, and have to perform just a new layer of encryption to the flying qubits that traverse the Qline. The conclusions and claims are supported by analytical proofs and - for the two-qubit example - by experimental data that are particularly impressive. The proposed methodology is sound, from both the analytical and experimental points of view. The organisation of the manuscript is good, in general, but may be further improved.

We thank the Reviewer for their careful analysis of our work and for acknowledging its originality, soundness, and significance to the related fields. Moreover, we thank the Reviewer for their precious suggestions aimed at improving our manuscript. We will now address point-by-point their comments.

1) Fig. 1 (inspired by Fig. 5 in [48]) requires reading [48] to be fully understood. In particular, the Server part in the figure is not clearly describe by the caption. Also, n is used to indicate the number of qubits that the client sends to the server, but later in the manuscript n is the number of clients. I would suggest to use m to indicate the number of qubits that the client sends to the server.

To better describe the roles of the client and the server in the BQC protocol:

- In the caption of Fig. 1, we now indicate by the letter “ m ” the number of qubits.
- We edited the caption of Fig. 1 as highlighted below (ref. [48] is now ref. [49]):

“In the preparation stage of the BQC protocol, a client randomly prepares m qubits and sends them to a quantum server. The server uses the qubits to form a resource state for the computation. In the measurement stage, $\mathcal{O}(m)$ rounds of classical communication between client and server are needed to carry out the computation. At each round, the server measures a qubit in a measurement basis suitably chosen by the client in order to hide the computation details, and it gives back the measurement outcome to the client. Then, the latter decides on the next measurement basis accordingly. Further detail about BQC is given in Protocol S4 of the Supplementary Information. Figure inspired by [49].”

2) The two-client protocol described in section II.A does not appear to be a 2-client instance of the multi-client protocol described in section II.B. More precisely, in the two-client protocol there is one source of entangled pairs that feeds Alice. In the multi-client scheme, there are m sources, each one distributing single qubits. This mismatch should be clarified, otherwise it seems misleading to state that the multi-client protocol is a “generalisation” of the two-client protocol. In my opinion, it would be better to start from the most general case, then illustrate the two-qubit example (which is the one that was experimentally implemented).

In our experimental implementation, we use a source of polarization-entangled photons for practical reasons. However, this does not change the hypotheses that our security and correctness proofs are built on, and it does not cause any loss of generality. Indeed, our architecture is in principle scalable, and, more importantly, the security proof we provide covers more general cases than the two-client protocol we implement, including the case of independent sources. Scaling to more qubits would require a server with full quantum capabilities, and, if this was the case, our protocol would be able to involve independent and arbitrarily distant qubit sources without any substantial change to its structure. In the organization of our manuscript, we preferred starting from the two-client because it represents a more complex case with respect to the original UBQC protocol, while a simpler case with respect to the multi-client scenario. Therefore, it seemed the best way, for easiness of understanding, to gradually arrive at the multi-client case. Moreover, most importantly, we believed it more suitable to present the two-client case first, since it helps introduce the experimental challenges leading to our design, and inspiring the extended protocol. To make these points of view clearer in our manuscript, we made the following changes:

- We added the following sentence in subsection IIB:

“Note that, here, we treat a more general case with respect to the previous one, hence the security proof we provide, which assumes a fully quantum server and independent and arbitrarily distant qubit sources, also holds for the implemented protocol.”

3) In the introduction, it is stated that “the quantum resource is first generated by a potentially untrusted server, then distributed to the clients”. However, in the manuscript, the potential unreliability of the quantum sources is not discussed. Actually, it seems that the quantum sources are always trusted, like the classical orchestrator. But while the orchestrator could be replaced by a classical SMPC protocol, it seems there is no way to replace the trusted sources. The authors should discuss this potential issue.

The point raised by the Reviewer is also strongly related to analogous observations by Reviewer 1. Let us also elaborate further on this point here. The protocol we present ensures that blindness is guaranteed even if the source of quantum states is not trusted. Indeed, we demonstrate that server S_1 , which provides quantum states, cannot get any information about input, output and computation details. Only when considering the correctness of the computation, the clients need to rely on the fact that the source generates the desired qubit states, as the problem of ensuring verifiability of BQC protocols based on remote state rotation still requires a full solution. To clarify this point, we made the following changes to the main text:

- In the “State preparation” paragraph of subsection IIA, we added the adjective ***untrusted*** when describing the source of maximally entangled states.
- We added the following sentence at the end of subsection IIC:

“We stress that the source of quantum states is not required to be trusted. Indeed, we demonstrated that none of the parties involved can gain any information about the input, output, and computation details.”

4) The discussion about the scalability of the protocol could be improved by describing how the approach based on a “fast electronic data elaboration circuit” (described in Supplementary Material S4) would scale with more than 2 clients.

We thank the Reviewer for allowing us to clarify the versatility of our experimental scheme and to give a concrete example of how to scale up our protocol to n clients. When clients are added to the protocol, each of them needs to apply a random rotation to the qubits involved in order to ensure blindness. Therefore, qubit rotation stations are added to the line for each client. Hence, the secret parameters involved in the protocol become:

$$\theta_i = \sum_j \theta_i^{C_j}$$

$$r_i = \bigoplus_j r_i^{C_j}$$

where $i = 1, 2$ is the qubit index while $j = 1, \dots, n$ is the client index. Analogous reasoning can be applied to input data x_i and algorithm details ϕ_i , whatever the distribution of such data among the clients. This change is easily handled by the circuit that embodies the TTP. Indeed, as shown in Supplementary Information S4, the circuit input A is computed from variables θ_2 , r_2 , and x_2 , which are computed by including the parameters of all clients. Parameter B , instead, simply takes the value of the algorithm ϕ_2 , which, once again, can be computed by summing up the values provided by the clients. To include this discussion in our main text, we implemented the following changes:

- To improve the “Scalability” paragraph in the “Discussion” section, as suggested by the Reviewer, we added the following sentence:

“On the other hand, from the protocol point of view, adding clients only requires adding rotation stations along each Qline, and our general proof technique covers the security aspects of such extensions. On the experimental side, instead, in our scheme, this would only imply handling more optical losses, which can be overcome with a brighter quantum source, and an accurate

characterization of the optical elements that perform the single-qubit rotations. This would not imply any substantial change even in the design of the fast data elaboration circuit, as we show in Supplementary Information section S4.”

- Furthermore, we added the following sentence in section S4 of the Supplementary Information:

“We now demonstrate that such a circuit would not need to be changed in its structure when having n clients. In our protocol, to ensure the privacy of all clients, each of them needs to rotate all the qubits involved in the computation. Then, one would first have to compute the cumulative secret rotation angle θ_i for the i -th qubit as $\theta_i = \sum_{j=1, \dots, n} \theta_i^{C_j}$ and the cumulative random bit $r_i = \bigoplus_{j=1, \dots, n} r_i^{C_j}$, where j indicates the j -th client. The classical input data x_2 and algorithm details ϕ_2 can be summed up analogously depending on the subset of clients that provided such information. Such cumulative parameters can then be placed in the coefficients A and B for the computation of the blind measurement angle δ_2 . The remainder of the computation would not change.”

The experimental methods are presented in a detailed fashion (the Supplementary Materials are quite rich), allowing for the reproduction of the work.

We thank the Reviewer for acknowledging the completeness of our Supplementary Material and the reproducibility of our experiment.

-
- [1] T. Kapourniotis, E. Kashefi, D. Leichtle, L. Music, and H. Ollivier, “Asymmetric quantum secure multi-party computation with weak clients against dishonest majority,” *arXiv preprint arXiv:2303.08865*, 2023.
 - [2] M. Doosti, L. Hanouz, A. Marin, E. Kashefi, and M. Kaplan, “Establishing shared secret keys on quantum line networks: protocol and security,” *arXiv preprint arXiv:2304.01881*, 2023.
 - [3] A. Broadbent, J. Fitzsimons, and E. Kashefi, “Universal blind quantum computation,” in *2009 50th Annual IEEE Symposium on Foundations of Computer Science*, pp. 517–526, IEEE, 2009.
 - [4] R.-T. Shan, X. Chen, and K.-G. Yuan, “Multi-party blind quantum computation protocol with mutual authentication in network,” *Science China Information Sciences*, vol. 64, pp. 1–14, 2021.
 - [5] G.-J. Qu and M.-M. Wang, “Secure multi-party quantum computation based on blind quantum computation,” *International Journal of Theoretical Physics*, vol. 60, pp. 3003–3012, 2021.
 - [6] M. Ciampi, A. Cojocaru, E. Kashefi, and A. Mantri, “Secure two-party quantum computation over classical channels,” *arXiv preprint arXiv:2010.07925*, 2020.
 - [7] E. Kashefi and A. Pappa, “Multiparty delegated quantum computing,” *Cryptography*, vol. 1, no. 2, p. 12, 2017.
 - [8] Y. Ma, E. Kashefi, M. Arapinis, K. Chakraborty, and M. Kaplan, “QEnclave – a practical solution for secure quantum cloud computing,” *npj Quantum Information*, vol. 8, no. 1, pp. 1–10, 2022.
 - [9] M. Hillery, V. Bužek, and A. Berthiaume, “Quantum secret sharing,” *Physical Review A*, vol. 59, no. 3, p. 1829, 1999.
 - [10] V. Dunjko, J. F. Fitzsimons, C. Portmann, and R. Renner, “Composable security of delegated quantum computation,” in *Advances in Cryptology – ASIACRYPT 2014* (P. Sarkar and T. Iwata, eds.), (Berlin, Heidelberg), pp. 406–425, Springer Berlin Heidelberg, 2014.

REVIEWERS' COMMENTS

Reviewer #1 (Remarks to the Author):

The authors have properly replied my questions. A minor comment for "proof-of-concept experimental demonstrations in different settings [26–34]" is that a related experiment shows the possibility of BQC over long distance is missing [Phys. Rev. Lett. 123, 100503 (2019)]. With this, I am happy to recommend its publication. The results are interesting and significant for general audience.

Reviewer #2 (Remarks to the Author):

The authors have made significant modifications according to all the comments and it can be accepted if the following minor problems are corrected.

1. In the title of Figure 1, " $O(m)$ rounds of classical communication between client and server are need to carry out the computation" should be changed as " $O(m)$ rounds of classical communication between a client and a server are need to carry out the computation"
2. Please uniformed the two notations Physical Review Letters and Physical review letters in the References.

Reviewer #3 (Remarks to the Author):

All the issues I pointed out have been carefully addressed. In general, I think the manuscript has further improved with respect to the initial submission. Therefore, I strongly support its publication.

Reply to Reviewer 1

The authors have properly replied my questions. A minor comment for "proof-of-concept experimental demonstrations in different settings [26–34]" is that a related experiment shows the possibility of BQC over long distance is missing [Phys. Rev. Lett. 123, 100503 (2019)]. With this, I am happy to recommend its publication. The results are interesting and significant for general audience.

We thank the Reviewer for their careful revision of our manuscript and for recommending its publication. In the revised manuscript, we added the reference mentioned by the Reviewer, which is now reference [27].

Reply to Reviewer 2

The authors have made significant modifications according to all the comments and it can be accepted if the following minor problems are corrected.

We thank the Reviewer for their careful revision of our work and for supporting its publication in this Journal. In the following, we address their comments point-by-point.

1. In the title of Figure 1, “ $\mathcal{O}(m)$ rounds of classical communication between client and server are need to carry out the computation” should be changed as ““ $\mathcal{O}(m)$ rounds of classical communication between a client and a server are need to carry out the computation””

We changed the caption of Fig. 1 as requested by the Reviewer. We report below the updated caption:

“In the preparation stage of the BQC protocol, a client randomly prepares m qubits and sends them to a quantum server. The server uses the qubits to form a resource state for the computation. In the measurement stage, $\mathcal{O}(m)$ rounds of classical communication between a client and a server are needed to carry out the computation. At each round, the server measures a qubit in a measurement basis suitably chosen by the client in order to hide the computation details, and it gives back the measurement outcome to the client. Then, the latter decides on the next measurement basis accordingly. Further detail about BQC is given in Supplementary Protocol 4. Figure inspired by [50].”

2. Please uniformed the two notations Physical Review Letters and Physical review letters in the References.

We now uniformed the two notations into Physical Review Letters.

Reply to Reviewer 3

All the issues I pointed out have been carefully addressed. In general, I think the manuscript has further improved with respect to the initial submission. Therefore, I strongly support its publication.

We thank the Reviewer for their careful revision of our manuscript and for supporting its publication in this Journal.